# The Effect of Local Cooling at the Elbow on Nerve Conduction Velocity and Motor Unit Behaviour: An Exploration of a Novel Neurological Assessment

**DOI:** 10.3390/s21206703

**Published:** 2021-10-09

**Authors:** Jim Richards, Antonin Gechev, Jill Alexander, Liane Macedo, Karen A. May, Steven B. Lindley

**Affiliations:** 1Allied Health Research Unit, University of Central Lancashire, Preston PR1 2HE, UK; JAlexander3@uclan.ac.uk; 2Department of Clinical Neurophysiology, Royal Free London Hospital, London NW3 2QG, UK; antonin.gechev@nhs.net; 3Faculty of Health Sciences of Trairí, Federal University of Rio Grande do Norte, Santa Cruz 59200-000, Brazil; liane.macedo@ufrn.br; 4School of Medicine, University of Central Lancashire, Preston PR1 2HE, UK; KAMay@uclan.ac.uk; 5Delsys Europe Ltd., Manchester M33 2DH, UK; slindley@delsyseurope.com

**Keywords:** motor unit firing rates, NCV, cooling, neurological assessments

## Abstract

Background: This study aimed to determine the effects of a standard therapeutic cooling protocol using crushed ice on the elbow to explore if changes in the motor unit (MU) firing rates in the first dorsal interosseous (FDI) muscle are comparable to known changes in sensory and motor nerve conduction velocity (NCV) due to a regional temperature drop around a peripheral nerve. Methods: Twelve healthy individuals were assessed before cooling, immediately after cooling, and 15 min of rewarming. Assessments included two standard non-invasive nerve conduction velocity tests and a non-invasive investigation of the MU firing rates using surface electromyography decomposition (dEMG). Results: Repeated ANOVAs showed significant differences in the MU firing rates and NCV between time points (*p* = 0.01 and *p* < 0.001). All measures showed significant differences between pre and post cooling and between pre-cooling and 15 min of passive re-warming, however, no changes were seen between post cooling and rewarming except in the sensory NCV, which increased but did not return to the pre-cooled state. Conclusions: This current study showed a significant, temporary, and reversible reduction in ulnar NCV across the elbow in healthy subjects, which was associated with a significant decrease in mean MU firing rates in the FDI muscle.

## 1. Introduction

Nerve conduction velocity (NCV) is a common assessment of peripheral nerve demyelinating conditions which is used in clinical practice. Focal slowing of peripheral conduction velocity is not believed to result in weakness but may account for a loss of deep tendon reflexes [1,2]. Nerve conduction slowing has been associated with a prolonged refractory period of transmission when a rapid train of impulses is transformed to a low rate train of impulses [3]. There are also well-known physiologic factors that can affect peripheral NCV such as sex, age, and temperature [4,5]. Temperature changes can exert a temporary effect mainly by altering action potential duration and its refractory period [5]. Henrikson [6] revealed that a 1-degree Celsius drop in temperature could cause a 2.4 m/s decrease in NCV, with conduction velocity decreasing by approximately 5% per degree Celsius [7,8]. The link between temperature and NCV has been extensively explored [9,10]. Recent studies have explored the physiological effects of cooling and report cold-induced impairment in muscle contractile properties, these were shown to result in neuromuscular compensatory mechanisms from altered motor unit (MU) firing properties [10]. The resulting peripheral dysfunction reported by Mallette et al. [11] (2018) has also been found to reduce manual dexterity [12] and can be associated with reductions in muscle strength [13].

Neuromuscular integrity in healthy subjects and individuals with different neurological conditions have been widely assessed using conventional nerve conduction studies (NCS) and electromyography (EMG) [11,14]. These concepts are based upon known functional responses, referred to as either spontaneous or provoked/stimulated EMG, where electrical signals pass along motor units (MUs) within the peripheral motor nervous system. Another technique routinely used is needle EMG, however, this invasive procedure has some risks and complications within NCS, despite being generally well tolerated, adverse events are sometimes reported that could limit patients’ response and can cause discomfort and bruising [15,16]. Therefore, the use of surface Electromyography (sEMG) has potential within neurological assessments to facilitate or augment existing assessment techniques.

Whilst the gross reaction of MUs to peripheral nerve axonal degeneration has been widely explored [17], little is known about the behaviour of individual MUs in peripheral nerve demyelination or cooling. Recently Mallette et al. [11] reported an increase in Motor Unit Action Potential (MUAP) duration and decreases in MUAP amplitude following cooling, measured from the decomposition of signals from surface Electromyography (dEMG). Innovative technologies and methods allowing the decomposition of signals from surface electromyography have been developed [18] allowing a more detailed non-invasive investigation of the action of individual MUs. Recent literature acknowledges the benefits of such technology to identify alterations in MU behaviour [19]. Subsequent investigations using dEMG have provided a greater insight into neuromuscular control in individuals with different neurological conditions, for example, Stroke [20], but to date, these have not been used diagnostically. Such additional detail could be useful in the detection of peripheral nerve demyelinating conditions such as acute or chronic inflammatory demyelinating neuropathies and may offer insights not possible with conventional NCS, which may be valuable in the detection of such conditions and the assessment of the efficacy of treatments. Therefore, this study aims to explore the effects of a standard therapeutic elbow cooling protocol on first dorsal interosseous (FDI) MU firing rates and a comparison with known changes in NCV in the proximal nerve segment, to explore the behaviour of individual MUs as a result of a regional temperature drop around a peripheral nerve.

## 2. Materials and Methods

### 2.1. Ethical Approval

This study was approved by the University of Central Lancashire ethics committee (STEMH 607) on the 10 March 2017 and conforms to the ethical standards of the Declaration of Helsinki. Participants were provided with information on the experimental protocol and associated risks before participation. Written and verbal consent were gained from all participants prior to data collection.

### 2.2. Participants

Twelve healthy individuals (7 males and 5 females) with an age of 30.1 ± 7.2 years, mass of 70.4 ± 8.6 kg, and height of 175 ± 8 cm were recruited from a university staff and student population. All participants were right hand dominant and met inclusion criteria, with no known disorders related to neuropathy, circulation or orthopaedic presentations.

### 2.3. Testing Procedure

Testing consisted of measurements taken with two standard non-invasive nerve conduction velocity tests and a non-invasive investigation of the MU firing rates using surface Electromyography decomposition [18]. The NCS testing was performed by an experienced trained clinical neurophysiologist, and the dEMG data collection was performed by experts in the technique. Participants had a standardised cooling protocol applied at the elbow joint using crushed ice and water (wetted ice) for 20 min, which is a standard method used in clinical practice for conditions including tennis elbow and throwers elbow [21]. This method has been shown to cool the skin to approximately 10 degrees Celsius, which is considered to be within a safe therapeutic range [22]. It has previously been shown that cooling at the skin significantly correlates with nerve temperature changes [23]. The cooling protocol was supervised by a trained allied health professional. The NCS and dEMG data were collected prior to cooling, immediately after the removal of cooling, and after 15 min of passive rewarming in a room with an ambient temperature of 20 degrees Celsius.

### 2.4. Nerve Conduction Velocity Tests

The supramaximal ulnar electrical nerve stimulation was performed using a previously reported technique [24]. Bar electrodes were placed over the wrist, below and above the elbow using a three-channel keypoint stimulator (Optima, Guildford, UK), with an interelectrode spacing of 4 cm. The distance between stimulating electrodes across the elbow was kept constant (10 cm) with the “below elbow” electrode positioned 4 cm distal to the olecranon and the “above elbow” electrode positioned 6 cm proximal to the olecranon. The upper limb position was kept constant during the experiment with the arm resting on a pillow with the elbow flexed at approximately 90°. The same stimulating sites and current intensities were used for both motor and sensory NCS (Figure 1).

The compound muscle action potentials (CMAPs) were recorded non-invasively from the right FDI muscle with disposable surface electrodes placed over the muscle belly and with the reference electrode placed over the metacarpophalangeal joint of the thumb [24], and the ground electrode placed on the dorsal surface of the hand. The ulnar motor nerve conduction velocity was calculated between the wrist and just below the elbow and across the elbow. In addition, antidromic sensory nerve action potentials (SAP) were recorded using ring electrodes placed over the 5th digit, with the active and reference electrodes placed 4 cm apart. The ulnar sensory nerve conduction velocity was calculated throughout the segments: wrist to 5th digit; wrist to below the elbow and across the elbow [24]. The motor and sensory potential latencies were taken from the onset of the negative peak (in ms) and the motor and sensory potential amplitudes were measured from the baseline to the negative peak (in mV and uV, respectively). The duration of the negative motor and sensory potential peak (in ms) was also measured. The same protocol was used whilst measuring the response obtained after ulnar nerve stimulation at the wrist, and below and above the elbow.

### 2.5. dEMG Data Collection

Surface EMG data were collected using the Delsys dEMG system (Delsys Inc., Natick, MA, USA). The dEMG system (Figure 2) consisted of a 16-channel amplifier and a 4-channel EMG electrode (dEMG sensor, Delsys Inc., Natick, MA, USA). The EMG signals were sampled at 2 kHz with a gain of 1000 and band-pass filtered (20–450 Hz) through a Delsys Bagnoli amplifier and NI USB 6251 DAQ (National Instruments, Austin, TX, USA). The dEMG sensor consisted of five cylindrical blunt-ended pins 0.5 mm in diameter, with an interelectrode spacing of 5 mm, providing 4 channels of surface EMG data. The dEMG sensor was taped in place over the surface of the FDI muscle after the skin was prepared by cleansing using an alcohol swab and moistening with a dilute saline solution. Initially, participants were asked to perform a maximum contraction pushing their index finger against a rigidly mounted force sensor for approximately 3 s; they were then allowed to rest for 2 min. They were then asked to perform 3 repetitions of a 30 s contraction at 30% of their maximum following a trapezoid force biofeedback channel. This comprised of a quiescent period, a 3 s ramp up period, a 30 s isometric hold, a 3 s ramp down period, and a final quiescent period with a 2 min rest between repetitions, the dEMG sensor remained attached to the subject for the duration of the experiment.

### 2.6. dEMG Data Analysis

The 4 channels of EMG recorded during each contraction were decomposed into their constituent motor unit action potential trains (MUAPTs) using the method previously described [18,25,26], which has been shown to have good intra- and inter-day reliability [27]. MUs with an accuracy greater than 90% were included in subsequent analysis, which was determined by the use of the decompose-synthesize-decompose-compare (DSDC) method introduced by Nawab et al. [26] and further developed by De Luca and Contessa [28]. The mean firing rates (MFR) of each MU during a single 30 s isometric contraction were exported and the overall mean, upper, middle and lower tertials for each contraction were found.

### 2.7. Statistical Analysis

All data were found to be normally distributed using the Shapiro–Wilk test and therefore found to be suitable for parametric statistical analysis. Repeated measures ANOVAs were used to compare the time points; pre-cooling, immediately post-cooling, and 15 min rewarming for the MU firing rates and nerve conduction velocity tests, followed by least significant difference post hoc pairwise comparisons where significant main effects were observed.

## 3. Results

The repeated measures (RM) ANOVAs showed significant differences in both motor and sensory ulnar nerve conduction velocity measures across the elbow (*p* < 0.001), and the sensory ulnar nerve conduction velocities between the wrist and 5th digit (*p* < 0.02) and wrist and below elbow (*p* = 0.048). However, no significant differences were seen in the motor ulnar nerve conduction velocities between below the elbow and wrist, Table 1, and no significant differences were seen in any of the CMAP or SAP amplitudes and durations at the different locations, Table 2; whereas, the MFR data showed significant differences between time points for the total mean firing rate (*p* = 0.01), and for the upper and middle tertials (*p* < 0.05), Table 3.

Post hoc pairwise comparisons showed significant differences across the elbow between pre-cooling and post cooling, between pre-cooling and 15 min of re-warming but not between post cooling and rewarming, with the exception of sensory NCV, which increased but did not return to the pre-cooled state. The ulnar nerve sensory conduction velocity showed small but significant changes between pre-cooling and 15 min rewarming in the wrist to 5th digit and wrist to below elbow. In addition, the upper and middle tertiles of the MU firing rates showed significant reductions between pre and post cooling, with the middle tertile showing a significant increase between post cooling and 15 min rewarming (*p* = 0.029) and a trend towards a significant difference in the upper tertile (*p* = 0.076) (Figure 3 and Table 4).

## 4. Discussion

This exploratory study investigated the effects of cooling at the elbow on ulnar NCV and FDI MU behaviour to explore if dEMG could be utilised for neurological assessments. A cooling application of wetted ice for a 20 min duration has been shown to produce a significant skin surface temperature reduction [29]. The application of wetted ice for 20 min on the elbow resulted in significant differences in the MFR between time points (*p* = 0.01) and NCS data (*p* < 0.001). Both MFR and NCV measures demonstrated significant differences, which could also be considered as clinically important changes, between pre-cooling and post cooling and between pre-cooling and 15 min of re-warming. It was also possible to observe significant differences between post cooling and rewarming for sensory NCV, which increased but did not return to the pre-cooled state. The middle and upper tertiles of the MU firing rates showed full recovery after 15 min of re-warming. The findings of the current study suggest the potential use of dEMG within clinical assessment may be of benefit when motor control is affected and NCV is difficult to measure in patient population groups with suspected peripheral neuropathy, affecting proximal limbs motor peripheral or cranial nerves, for example, long thoracic; axillary; musculocutaneous; anterior interosseous; obturator; gluteal; spinal accessory; facial, etc., or to facilitate or augment existing assessment techniques, and may also help to reduce some of the risks and complications which have been reported with needle EMG in a small number of cases [15,16].

A recent study exploring motor unit properties following muscle cooling at the elbow by Mallette et al. [11] reported increases in the number of MUs recruited, detected through surface dEMG. The relationship between MU firing rate and the threshold of recruitment changed, suggesting that after cooling the MU recruitment was earlier and/or reached higher firing rates; however, mean motor unit firing rate and recruitment threshold did not show any significant changes between neutral and cold conditions [11]. This is consistent with Cornwall [30] who reported a neural strategy that compensates for muscle impairment associated with cooling, providing a rationale for why recruitment of MUs in the early phase of a contraction appear to present higher firing rates during cold exposure. In contrast, MUs recruited later during muscle contraction following cooling demonstrated lower firing rates in comparison to neutral temperatures [10].

The findings of this current study are supported by Mallette et al. [11] who showed a significant, 30% temporary and reversible reduction in ulnar NCV across the elbow in healthy subjects, which was associated with a significant, 15% decrease, in mean MU firing rate in the FDI muscle. This provides evidence to support a direct electrophysiological motor deficit caused solely by focal nerve conduction slowing. The implications of a reduced MU MFR include a reduction in manual dexterity [12] and joint control accuracy [28]. Furthermore, reductions in skin surface sensation following cooling suggest an effect on feedback which may be linked to a reduced sensory NCV [31].

Since no proximal CMAP amplitude decay was seen after cooling these results rule out a classical nerve conduction block across the cooled ulnar nerve region at the elbow. It is known that hyperpolarisation and a reduced safety factor in demyelinating nerve segments may be demonstrated after high-frequency receptive nerves simulation or after sustained volitional activity with subsequent electrical stimulation [32]. Since in both cases a supramaximal electrical stimulation is necessary it could be hypothesised the assessment of rate-dependent conduction block is mainly focused on large nerve axons. The conventional view is that electrical stimulation progressively activates those neurones which may be associated with the innervation of larger, lower firing rate MUs.

A distinguishable feature of our study is that the first recruited MUs, firing with higher rates before nerve cooling, show a more pronounced firing rate drop in comparison with the later recruited MUs, indicating that the latter are less susceptible to a slowing of the nerve conduction velocity. It is unlikely a conduction block will cause an increase in the MU firing rates by itself, which might indicate other mechanisms within MU regulation and control. A potential application of the methods presented is in the assessment of plasticity of skeletal muscles [33], particularly in cases of altered MU firing rates due to the slowing of nerve conduction. These may also be useful in the assessment of treatment of hyperpolarisation-induced axonal dysfunction, either pharmacologically [34] or after applying a polarizing current over the affected nerve segment [35], which aims to prevent the development of a nerve conduction block. Therefore, clinically suspected internodal conduction failure and/or nerve conduction disturbance in regional nerve cooling could be verified indirectly and non-invasively using dEMG in superficially located muscles distal to the nerve involvement site. This may provide an alternative assessment when conventional NCS or high-frequency repetitive nerve stimulations are difficult to perform or are normal or inconclusive. Moreover, cooling is well known to improve conduction in demyelinating conditions [3], therefore, knowing the cooling response of healthy axons on the MUs firing rates would be recommended before studying it on diseased nerves.

Our study did not involve external electrical stimuli to provoke additional axonal conduction failure at the cooled nerve segment. The analysis involved volitionally activated MUs recorded over a muscle away from the direct site of the cooling. In addition, none of the participants experienced any cooling sensations of their hands during the protocol. Thus, the resulting mean MU firing rate drop after cooling would reflect an axonal conduction insufficiency for the whole recordable spectrum of motor neurons innervating the muscle. Although muscle temperature could not be monitored in this study, the lack of significant changes in motor and sensory nerve potentials amplitudes and duration throughout the whole assessment would rule out temperature inconsistencies at the level of the muscle.

Conduction slowing alone is the most common finding in acute and chronic, focal or generalised demyelinating conditions. Whilst its clinical significance in localising the nerve injury is well known, the impact of nerve conduction slowing on the muscle dysfunction, e.g., weakness and fatigue are rarely considered [36]. The clinical importance of a conduction block usually overcomes that of conduction slowing alone. It is difficult, however, to interpolate our findings with previous studies since no similar measurements to date are available for comparison in clinical practice. The closest published observations are in the field of entrapment neuropathies of the ulnar nerve [37], median nerve [38], or in chronic inflammatory demyelinating polyneuropathies [32,39]. Further studies are, therefore, needed to explore this technique in the assessment of demyelinating peripheral focal and generalised neuropathies.

## 5. Conclusions

This study identified a previously unreported association between the motor and sensory NCV and the mean MU firing rates. These findings demonstrate a nerve conduction-dependent fall in MU firing rates which may be associated with motor deficits, most likely due to the increased axonal refractory period after regional cooling. Furthermore, our results revealed that first recruited MUs are affected more than those recruited later when focal nerve conduction slowing is evident across a peripheral nerve. This provides further evidence for the potential of dEMG as a novel neurological assessment that could be used when motor conduction velocity is difficult to measure or to facilitate or augment existing assessment techniques. This would be especially useful early in the clinical state before axonal degeneration is evident, which is still challenging for routine neurophysiological methods.

## Figures and Tables

**Figure 1 sensors-21-06703-f001:**
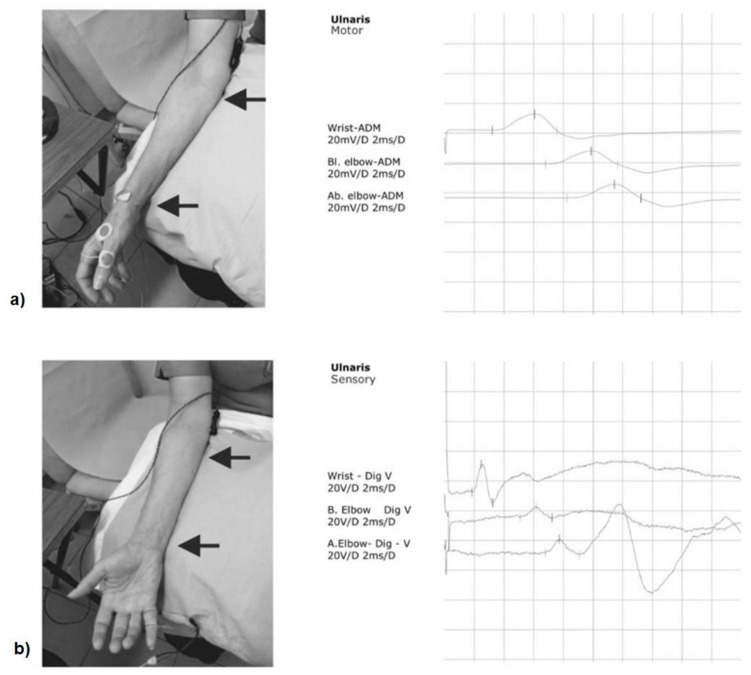
Ulnar nerve conduction velocity: (**a**) motor; (**b**) sensory.

**Figure 2 sensors-21-06703-f002:**
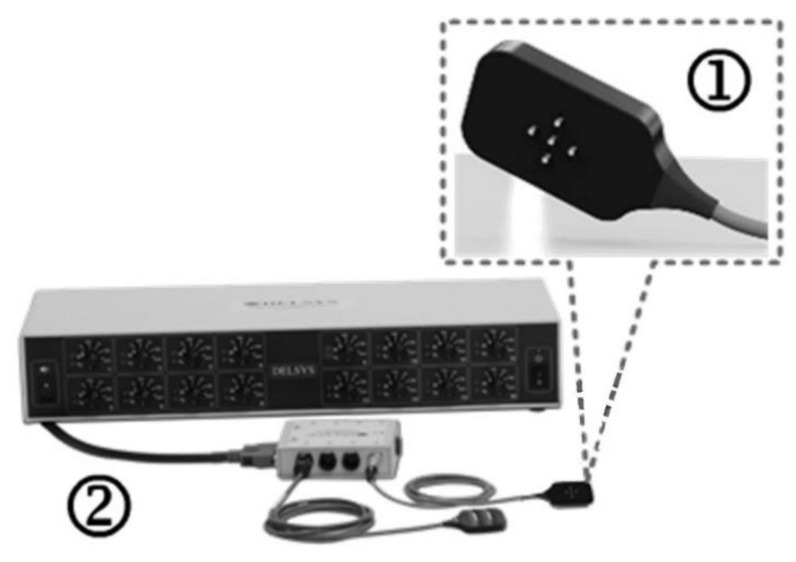
dEMG sensor (**1**); Delsys Bagnoli amplifier (**2**).

**Figure 3 sensors-21-06703-f003:**
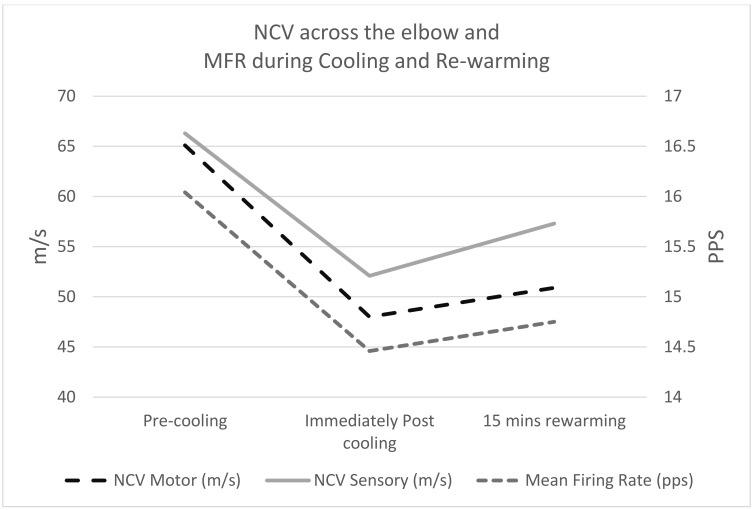
NCS and MFR during cooling and re-warming.

**Table 1 sensors-21-06703-t001:** Descriptive statistics and RM ANOVA for NCV.

	Motor NCV (m/s) Group Mean (SD)		Sensory NCV (m/s) Group Mean (SD)		
	Below Elbow to Wrist	Across Elbow	Wrist to 5th Digit	Below Elbow to Wrist	Across Elbow
Pre-cooling	63.3 (5.1)	65.1 (6.7)	58.1 (5.0)	63.9 (4.7)	66.3 (6.2)
Immediately Post cooling	62.3 (5.7)	48.0 (8.1)	56.4(6.6)	62.6 (6.0)	52.1 (7.2)
15 min rewarming	61.9 (6.5)	50.9 (8.6)	55.1 (5.4)	61.4 (6.1)	57.3 (8.2)
RM ANOVA	*p* = 0.085	*p* < 0.001	*p* = 0.020	*p* = 0.048	*p* < 0.001

NCV: nerve conduction velocity; SD: standard deviation.

**Table 2 sensors-21-06703-t002:** Descriptive statistics and RM ANOVA for CMAP or SAP amplitudes and durations.

	Wrist	Below Elbow	Above Elbow	Wrist	Below Elbow	Above Elbow
	CMAP Amplitude (mV) Group mean (SD)			SAP Amplitude (uV) Group mean (SD)		
Pre-cooling	10.2 (1.8)	9.6 (1.8)	9.0 (2.0)	26.2 (5.0)	12.2 (6.3)	11.7 (5.8)
Immediately Post cooling	9.9 (2.0)	9.6 (2.1)	9.0 (2.3)	26.5 (6.0)	13.7 (7.9)	12.4 (6.8)
15 min rewarming	9.9 (2.8)	9.4 (2.6)	9.1 (2.6)	27.5 (6.5)	13.6 (7.5)	12.8 (6.9)
RM ANOVA	*p* = 0.785	*p* = 0.835	*p* = 0.925	*p* = 0.606	*p* = 0.739	*p* = 0.843
	CMAP Duration (ms) Group mean (SD)			SAP Duration (ms) Group mean (SD)		
Pre-cooling	4.4 (0.6)	4.5 (0.6)	4.7 (0.6)	2.3 (0.4)	2.6 (0.6)	2.4 (0.5)
Immediately Post cooling	4.3 (0.6)	4.5 (0.7)	4.5 (0.7)	2.2(0.4)	2.7 (0.6)	2.5 (0.4)
15 min rewarming	4.5 (0.7)	4.5(0.7)	4.6 (0.6)	2.1 (0.4)	2.8 (0.6)	2.4 (0.5)
RM ANOVA	*p* = 0.764	*p* = 0.700	*p* = 0.179	*p* = 0.144	*p* = 0.529	*p* = 0.600

CMAP: compound muscle action potential; SAP: sensory nerve action potentials; SD: standard deviation; RM: repeated measure.

**Table 3 sensors-21-06703-t003:** Descriptive statistics and RM ANOVA for MFR.

		Mean (SD)		
	Total Mean Firing Rate (pps)	Upper Tertile Firing Rate (pps)	Middle Tertile Firing Rate (pps)	Lower Tertile Firing Rate (pps)
Pre-cooling	16.04 (2.59)	20.79 (2.26)	15.84 (2.01)	11.35 (1.71)
Immediately Post cooling	14.46 (2.51)	18.78 (2.49)	14.42 (1.88)	10.45 (1.42)
15 min rewarming	14.75 (2.24)	20.59 (2.45)	16.02 (1.62)	11.23 (1.22)
RM ANOVA	*p* = 0.01	*p* = 0.038	*p* = 0.038	*p* = 0.247

pps: pulses per s; RM: repeated measure.

**Table 4 sensors-21-06703-t004:** Post hoc pairwise comparisons.

	Mean Difference	% Change	Upper and Lower CI of the Difference	*p*-Value
Motor NCV Across Elbow				
Pre-cooling vs. post cooling	**17.1** *	26%	12.1 to 22.2	**<0.001**
Pre-cooling vs. 15 min rewarming	**14.2** *	22%	9.4 to 19.0	**<0.001**
Post cooling vs. 15 min rewarming	−3.0	−6%	−6.6 to 0.675	0.101
Sensory NCV wrist to 5th digit				
Pre-cooling vs. post cooling	1.76	3%	−0.08 to 3.60	0.059
Pre-cooling vs. 15 min rewarming	**3.02** *	5%	0.40 to 5.65	**0.028**
Post cooling vs. 15 min rewarming	1.27	2%	−0.74 to 3.27	0.192
Sensory NCV wrist to below elbow				
Pre-cooling vs. post cooling	1.32	2%	−1.13 to 3.77	0.262
Pre-cooling vs. 15 min rewarming	**2.52** *	4%	0.29 to 4.74	**0.030**
Post cooling vs. 15 min rewarming	1.20	2%	−0.31 to 2.70	0.108
Sensory NCV across elbow				
Pre-cooling vs. post cooling	**14.2** *	21%	10.1 to 18.3	**<0.001**
Pre-cooling vs. 15 min rewarming	**9.0** *	14%	4.937 to 13.0	**<0.001**
Post cooling vs. 15 min rewarming	**−5.2** *	−10%	−8.1 to −2.4	**0.002**
Total mean firing rate (pps)				
Pre-cooling vs. post cooling	**1.58** *	9%	0.81 to 2.36	**<0.001**
Pre-cooling vs. 15 min rewarming	**1.29** *	8%	0.50 to 2.08	**0.002**
Post cooling vs. 15 min rewarming	−0.29	−2%	−1.06 to 0.48	0.454
Upper tertile firing rate (pps)				
Pre-cooling vs. post cooling	**2.02** *	10%	0.27 to 3.76	**0.029**
Pre-cooling vs. 15 min rewarming	0.20	1%	−1.36 to 1.77	0.771
Post cooling vs. 15 min rewarming	−1.81	−10%	−3.86 to 0.23	0.076
Middle tertile firing rate (pps)				
Pre-cooling vs. post cooling	**1.43** *	9%	0.05 to 2.81	**0.044**
Pre-cooling vs. 15 min rewarming	−0.17	1%	−1.68 to 1.33	0.789
Post cooling vs. 15 min rewarming	**−1.60** *	−10%	−2.99 to −0.22	**0.029**

NCV: nerve conduction velocity; pps: pulses per s; CI: confidence interval.

## Data Availability

Not available.

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
