# Peer review of "The Effect of Local Cooling at the Elbow on Nerve Conduction Velocity and Motor Unit Behaviour: An Exploration of a Novel Neurological Assessment"

_sensors, 2021, doi:10.3390/s21206703_

Round 1
Reviewer 1 Report
In this manuscript, the authors examine surface detected motor unit firing rates in ulnar innervated muscles before, during, and after cooling at the elbow. This is an interesting approach. The manuscript would benefit from addressing the following points:
- The study would be much stronger if a few, or even one, patient with ulnar neuropathy were included. There is no indication from the current study that those with ulnar neuropathy will be substantially different from healthy subjects.
- It's hard to imagine that these techniques will replace needle EMG. Needle EMG has advantages of detecting denervation, reinnervation, muscle fibrosis and other findings that guide clinical treatment. The risks and complications of needle EMG are relatively small. Doubt a surgeon would operate based solely on these studies. This study might be better framed as a way to understand motor unit physiology rather than as a potential replacement for needle EMG, especially if one doesn't have patients to compare.
- While it might not be realistic to collect this data now, it would have been interesting to collect strength or dexterity data during cooling. Maybe for the limitations section.
- While clearly there was nerve cooling, as the NCV slowed, it might be reasonable to mention that surface skin temperature is not the same as nerve temperature.
- It would be interesting to consider the hypothesis that those with demyelinating ulnar neuropathy may show less difference with cooling. Since cooling prolongs depolarization times, it actually helps to overcome conduction block -e.g. MS patients who are better in the cold.
- The authors report " The findings of the current study suggest the potential use of dEMG within clinical assessment may be of benefit when motor control is affected and NCV is difficult to measure in patient population groups with suspected peripheral neuropathy, or may help to reduce some of the risks and complications which have been reported with needle EMG." It's hard imagine a situation in which NCV cannot be measured and where dEMG would be useful. Usually it's only in very severe lesions that one can't obtain a CMAP.
- Minor points: please spell out pps for first usage. Also, there is some repetition in the tables and the figure.
Author Response
In this manuscript, the authors examine surface detected motor unit firing rates in ulnar innervated muscles before, during, and after cooling at the elbow. This is an interesting approach. The manuscript would benefit from addressing the following points:
- The study would be much stronger if a few, or even one, patient with ulnar neuropathy were included. There is no indication from the current study that those with ulnar neuropathy will be substantially different from healthy subjects.
Response: Thank you for the suggestion, and we agree that further exploring these methods on a patient population would be compelling and the natural progression of the work. The aim of this work was to explore these novel techniques on a healthy population, to then allow refinement before applying to a patient population. - It's hard to imagine that these techniques will replace needle EMG. Needle EMG has advantages of detecting denervation, reinnervation, muscle fibrosis and other findings that guide clinical treatment. The risks and complications of needle EMG are relatively small. Doubt a surgeon would operate based solely on these studies. This study might be better framed as a way to understand motor unit physiology rather than as a potential replacement for needle EMG, especially if one doesn't have patients to compare.
Response: We agree, this novel technique is suggestive to facilitate or augment existing assessment techniques rather than replace. We have clarified this on lines 60 in the introduction and line 265 in the discussion, and highlighted that risks and complication only occur in a small number of cases on line 267. - While it might not be realistic to collect this data now, it would have been interesting to collect strength or dexterity data during cooling. Maybe for the limitations section.
Response: We agree that this additional data could supplement our existing explorations and will consider this for future study designs. - While clearly there was nerve cooling, as the NCV slowed, it might be reasonable to mention that surface skin temperature is not the same as nerve temperature.
Response: We have added a reference to support the correlation between skin and nerve temperatures. See lines 110-111 - It would be interesting to consider the hypothesis that those with demyelinating ulnar neuropathy may show less difference with cooling. Since cooling prolongs depolarization times, it actually helps to overcome conduction block -e.g. MS patients who are better in the cold.
Response: Thank you again for the suggestion, and we agree that further exploring these methods on a patient population, and agree people with MS would be an interesting group to consider, in particular when considering responses to treatment. - The authors report " The findings of the current study suggest the potential use of dEMG within clinical assessment may be of benefit when motor control is affected and NCV is difficult to measure in patient population groups with suspected peripheral neuropathy, or may help to reduce some of the risks and complications which have been reported with needle EMG." It's hard imagine a situation in which NCV cannot be measured and where dEMG would be useful. Usually it's only in very severe lesions that one can't obtain a CMAP.
Response: There are situations where NCS is not routinely measured for the proximal peripheral nerves or cranial nerves for example: Spinal Accessory, Facial, Long Thoracic; Axillary, Musculocutanoues, Anterior interroseus, and Gluteal
We have added this to the text on lines: 251-253
“.. suspected peripheral neuropathy affecting proximal limbs peripheral or cranial nerves: for example Spinal Accessory; Axillary; Musculocutanoues; Anterior interroseus; Obturator; Gluteal; Facial etc. ...”
- Minor points: please spell out pps for first usage. Also, there is some repetition in the tables and the figure.
Response: Sorry for this oversight now corrected.
Reviewer 2 Report
- Authors utilized Shapiro–Wilk test stating that it is suitable for parametric statistical analysis. Is there a rationale behind this decision apart from that being suitable. For instance, is there another example in the literature? What are other methods that can be employed? Are there other similar studies employed Shapiro-Wilk test? Can authors investigate and cite the relevant papers?
- There are major and important results in this study. However, authors have used very long and complicated sentences throughout the discussion. For instance, please see the sentence below from the text (between lines of 238 and 243)
"MFR and both NCV measures demonstrated significant differences, which 238 could also be considered as clinically important changes, between pre-cooling and post 239 cooling and between pre-cooling and 15 minutes of re-warming, and between post cooling 240 and rewarming for sensory NCV, which increased but did not return to the pre-cooled 241 state, and the middle and upper tertiles of the MU firing rates which showed full recovery 242 after 15 minutes of re-warming."
I must state that this is very long sentence and also very complicated. It ıs very difficult to extract the information. Unfortunately, this is not the only example. Please scan the paper and shorten such load sentences and make them clear. - Authors presented results in tables with using standard deviation. However, the number of repeats are not mentioned in the results section. Does this mean come from the one measurement of 12 samples? If not, how many repeats were conducted? Can you please clearly mention where the "mean" comes from in the results section?
- Authors state that (lines: 125-129)
"The Compound Muscle Action Potentials (CMAPs) were recorded non-invasively 125 from the right First Dorsal Interosseous (FDI) muscle with disposable surface electrodes 126 placed over the muscle belly and with the reference electrode placed over the metacar- 127 pophalangeal joint of the thumb, and the ground electrode placed on the dorsal surface of 128 the hand. The Ulnar Motor Nerve Conduction Velocity was calculated between the wrist 129 and just below the elbow and across the elbow."
Why did they place the electrodes in such a way? Was this choice coming from an example in the literature? Or did they try different configurations and decide on this one? I am asking this because the placing of the electrodes changes the results dramatically.
Author Response
- Authors utilized Shapiro–Wilk test stating that it is suitable for parametric statistical analysis. Is there a rationale behind this decision apart from that being suitable. For instance, is there another example in the literature? What are other methods that can be employed? Are there other similar studies employed Shapiro-Wilk test? Can authors investigate and cite the relevant papers?
Response: The Shapiro–Wilk test is one of two very common options alongside the Kolmogorov-Smirnov test to determine if the data is normally distributed.
All data were found to be normally distributed using the Shapiro–Wilk test and therefore parametric statistical analysis is recommended.
If the Shapiro–Wilk tests found the data to be not normally distributed we would have used non-parametric statistics i.e. Freidman tests with post hoc Wilcoxon-Signed Rank test to explore the comparisons between time intervals.
- There are major and important results in this study. However, authors have used very long and complicated sentences throughout the discussion. For instance, please see the sentence below from the text (between lines of 238 and 243)
"MFR and both NCV measures demonstrated significant differences, which 238 could also be considered as clinically important changes, between pre-cooling and post 239 cooling and between pre-cooling and 15 minutes of re-warming, and between post cooling 240 and rewarming for sensory NCV, which increased but did not return to the pre-cooled 241 state, and the middle and upper tertiles of the MU firing rates which showed full recovery 242 after 15 minutes of re-warming."
I must state that this is very long sentence and also very complicated. It ıs very difficult to extract the information. Unfortunately, this is not the only example. Please scan the paper and shorten such load sentences and make them clear.
Response: Thank you for your comment. We have reviewed the manuscript and edited to try and make the sentences shorter where possible to improve the readability.
- Authors presented results in tables with using standard deviation. However, the number of repeats are not mentioned in the results section. Does this mean come from the one measurement of 12 samples? If not, how many repeats were conducted? Can you please clearly mention where the "mean" comes from in the results section?
Response: We have added “a single” 30 second isometric contraction” to line 186. It was not possible to take more readings as rewarming was occurring. The mean and standard deviations relate to the grouped mean and sd from all participants. We have added “Group Mean (sd)” to table 1 and 2.
- Authors state that (lines: 125-129)
"The Compound Muscle Action Potentials (CMAPs) were recorded non-invasively 125 from the right First Dorsal Interosseous (FDI) muscle with disposable surface electrodes 126 placed over the muscle belly and with the reference electrode placed over the metacar- 127 pophalangeal joint of the thumb, and the ground electrode placed on the dorsal surface of 128 the hand. The Ulnar Motor Nerve Conduction Velocity was calculated between the wrist 129 and just below the elbow and across the elbow."
Why did they place the electrodes in such a way? Was this choice coming from an example in the literature? Or did they try different configurations and decide on this one? I am asking this because the placing of the electrodes changes the results dramatically.
Response: This is a standard configuration used in clinical neurophysiological assessments. We have added a reference that supports the use of this technique
“The supramaximal ulnar electrical nerve stimulation was performed using a previously reported technique [23].”
Reviewer 3 Report
Some references are very old; from 1951, 1956, 1973, 1977. The reviewer suggests to replace these publications with more recent studies
Is the number of twelve individuals included in the study sufficient from the point of view of statistics? It seems too small.
The novelty of study should be more clearly emphasized in conclusion
The list of abbreviations used in the manuscript should be added to the beginning of the manuscript
All abbreviations must be explained at the time of first citation in the text (for example FDI – line 18)
The number and date of the ethical consent should be added
Author Response
Reviewer 3
Open ReviewSuggestions for Authors
Some references are very old; from 1951, 1956, 1973, 1977. The reviewer suggests to replace these publications with more recent studies
Response: We have replace some of the more dated references where more current papers exist.
Is the number of twelve individuals included in the study sufficient from the point of view of statistics? It seems too small.
Response: We take the reviewer’s point so we made sure that the title reflected that this is exploratory. We have further added at the start of the discussion
“This exploratory study..”
and at the end of the introduction
“Therefore this study aims to explore the effects…”
- The novelty of study should be more clearly emphasized in conclusion
Response: Thank you for this comment. We have added these statements to the conclusion
“identified a previously unreported association..”
and
“This provides further evidence for the potential of dEMG as a novel neurological …”
“facilitate or augment existing assessment techniques”
- The list of abbreviations used in the manuscript should be added to the beginning of the manuscript
All abbreviations must be explained at the time of first citation in the text (for example FDI – line 18)
Response: now added, we apologize for this oversight
The number and date of the ethical consent should be added
Response: now added, we apologize for this oversight